# A Supervoxel-Based Random Forest Method for Robust and Effective Airborne LiDAR Point Cloud Classification

Lingfeng Liao [1], Shengjun Tang [1,*], Jianghai Liao [1], Xiaoming Li [1], Weixi Wang [1], Yaxin Li [2] and Renzhong Guo [1]

1   School of Architecture and Urban Planning, Research Institute for Smart Cities, Key Laboratory of Urban Land Resources Monitoring and Simulation, Ministry of Natural Resources, Shenzhen University, Shenzhen 518060, China; 2018104003@email.szu.edu.cn (L.L.); liaojh@szu.edu.cn (J.L.); lixming@szu.edu.cn (X.L.); wangwx@szu.edu.cn (W.W.); guorz@szu.edu.cn (R.G.)
2   Shenzhen Research Institute, The Hong Kong Polytechnic University, Shenzhen 518057, China; yaxin-1.li@polyu.edu.hk
*   Correspondence: shengjuntang@szu.edu.cn; Tel.: +86-18665363527; Fax: +86-755-2653-0210

**Abstract:** As an essential part of point cloud processing, autonomous classification is conventionally used in various multifaceted scenes and non-regular point distributions. State-of-the-art point cloud classification methods mostly process raw point clouds, using a single point as the basic unit and calculating point cloud features by searching local neighbors via the k-neighborhood method. Such methods tend to be computationally inefficient and have difficulty obtaining accurate feature descriptions due to inappropriate neighborhood selection. In this paper, we propose a robust and effective point cloud classification approach that integrates point cloud supervoxels and their locally convex connected patches into a random forest classifier, which effectively improves the point cloud feature calculation accuracy and reduces the computational cost. Considering the different types of point cloud feature descriptions, we divide features into three categories (point-based, eigen-based, and grid-based) and accordingly design three distinct feature calculation strategies to improve feature reliability. Two International Society of Photogrammetry and Remote Sensing benchmark tests show that the proposed method achieves state-of-the-art performance, with average F1-scores of 89.16 and 83.58, respectively. The successful classification of point clouds with great variation in elevation also demonstrates the reliability of the proposed method in challenging scenes.

**Keywords:** point cloud classification; supervoxel; random forest; feature fusion; segmentation

## 1. Introduction

With the development of photogrammetry and light detection and ranging (LiDAR) technologies, urban three-dimensional (3D) point clouds can be easily obtained. Three-dimensional point cloud data are used in many applications, such as power line inspections [1], urban 3D modeling [2,3], and unmanned vehicles [4]. However, the most basic requirement for these applications is the semantic classification of 3D point cloud data, which has been a research focus among photogrammetry and remote sensing communities.

Early classification efforts mainly focused on extracting low-level geometric primitives, such as point features, line features, and surface features, which were used for surface reconstruction or point cloud alignment. In recent years, researchers have developed methods for extracting high-level semantic features for structure model reconstruction from point cloud data through machine learning-and deep learning-based methods [5–7]. The core challenges of point cloud data classification are extracting discriminative features from neighborhoods and constructing point cloud classifiers [8,9]. Accurate classification depends on a combination of robust point cloud features and proper classifiers [8,10]. Recent works have applied deep learning networks to directly learn per-point features from raw point clouds [11,12]. Similar to traditional machine learning, these methods

focus on the extraction of higher-order features from point cloud data by building a new convolutional neural network. Although remarkable performance has been achieved using these methods, large training sample sets are required to pre-train the classification models. These semantic tags require manual labeling, which is time-consuming and labor-intensive. Moreover, the training models obtained by such methods are difficult to generalize to other scenarios [13].

To solve the model generalization and incomplete label data problems, many researchers prefer traditional machine learning methods, which require only a small sample dataset to achieve fast and accurate semantic point cloud data classification [14–16]. However, original point cloud features are often highly unstable due to the influence of point cloud data accuracy and noise, especially data acquired by tilt photogrammetry. Thus, more researchers are exploiting high-order features and their contextual information for scene classification. As dimensional objects expanding upon the concept of the "superpixel" [17], "supervoxels" [18] are generated by partitioning 3D space as point clusters. Supervoxels have been increasingly applied to describe adjoining points related to the same objects [16,19]. Transferring the original point cloud to the "supervoxel cloud" propagates simple point-based classification to an object-based level. Some point cloud segmentation methods, such as locally convex connected patches (LCCP), recognize points through supervoxel-adjacent relationships. In addition to features, classifiers that can effectively deal with massive data must be considered. Machine learning methods such as random forest (RF) that are capable of handling complex data are gaining attention for this purpose [20,21].

Here, we propose a robust and effective point cloud classification approach that integrates point cloud supervoxels and their LCCP relationships into an RF classifier. The proposed method involves three strategies to effectively improve classification accuracy. (1) Features are divided into three categories based on their description types (point-based, eigen-based, and grid-based), and three unique feature calculation strategies are designed to improve feature reliability. (2) A centroid point is used to represent supervoxel geometries, and every point that belongs to the same cluster shares all properties. (3) Supervoxel local neighborhoods are segmented by LCCP to avoid the inclusion of object borders.

The rest of this paper is organized into four sections. In Section 2, we review and compare similar methods for solving classification issues in two categories. Section 3 presents the framework of the proposed supervoxel-based RF model, providing the feature descriptions and RF model process and algorithm. The statistical and visual results of the data training and validation are shown in Section 4, and our research conclusion and remarks are given in Section 5.

## 2. Related Works

Previous classification approaches can be categorized as knowledge-driven and model-driven methods predicated on the classifier type. Reviews of the logical bases for these methods are presented below.

### 2.1. Knowledge-Driven Methods

Knowledge-driven methods involve the detection of structural features consisting of points; human expert knowledge of the terrestrial surface is then used to extract various objects from the original point cloud. In some cases, correction systems are applied to fix obvious faults [16]. Typically, these approaches focus on two crucial points: what features to extract and how to build a reliable human-knowledge-based system for classification. Generally, some human-eye optical features, such as height, slope, and color, can be used in real cases. Huang et al. [22] integrated multispectral imagery and ALS data to obtain the ground truth red–green–blue (RGB) color and surface elevation values in each pixel and built a classification system based on color information and urban elevation knowledge for executing segmentation of different objects. Germaine and Hung [23] constructed two systems based on surface height and surface slope, respectively. Polygonal features can also

be used in knowledge-based approaches. For instance, Zheng et al. [24] used the Fourier fitting method [25] to classify the pointcloud, in which the geometrical eigen features and basic features were integrated in their classification algorithm. Including spectral information assures reliable results, and combining various features ensures the system has high performance. Additionally, simple rules derived from gained features facilitate increased accuracy in the postprocessing stage. By regularizing objects placed at different heights and with distinctive surface slopes, a correction system can fix local classification faults in point clouds [22,26].

Knowledge-driven methods are well-acknowledged for their succinct and distinct processes based on the human recognition of ground objects [26]. However, these approaches rely on prior information, and precise airborne imagery is essential for acquiring reliable outputs. Moreover, matching the LiDAR dataset and multispectral image coordinates is time-consuming, which restricts knowledge-based processes to a small area range and can create spectral error accumulation. Furthermore, specific knowledge cannot generalize to diverse situations, such as vehicles and clusters on a small scale, which may generate errors in the final output. Thus, complex urban scenes may be challenging to classify using knowledge-driven methods.

### 2.2. Model-Driven Methods

Model-driven methods construct classification models from features extracted from or calculated based on point clouds, before segregating clouds into a training dataset and validation dataset. The training set fits the model and modifies the original parameters, and the validation set provides the current classification performance of the model. Appropriate model structures are crucial for such methods. The primary differences between knowledge- and model-driven methods are the classifier types and structures.

Many approaches use convolutional neural networks [27] as the basic model structure [28–31]. The network structure is designed according to the actual composition of the point cloud dataset, and then the points are separated into clusters used for input. Through many rounds of forward and backward propagation, a relatively reliable classification model can be built. Varied features are included to increase the input complexity and optimize model performance. Wang et al. [31] developed a dynamic graph network structure that could simultaneously finish classification and segmentation to identify shape properties and include neighborhood features. Hong et al. [28] built upon this method by including a modification module to balance the performance and cost and using an optimized skip connection network for efficient training. Classic models, such as RF, conditional random fields [14] with integrated RF, and support vector machines [32], have also been used for the labeling process [21,33,34]. The supervoxel-based method representing object-based routes has been incorporated into simple classifiers [35], and the supervoxel-adjacency relationship can also be considered as a feature of the local neighborhood [36].

Most existing model-driven methods based on supervoxel extraction are prone to include real object boundaries in the local neighborhood of voxels, which decreases the homogeneity of supervoxel adjacency and polygonal feature accuracy. Combining a precise object segmentation utility with previous model-driven methods will effectively solve this problem. Object edges can be detected by particular network structures or LCCP [37]. Feng et al. [38] developed a local attention-edge convolutional network that identified objects by summarizing the features of all neighbors as a weight value learned by the network. The LCCP examined the connection between two adjacent supervoxels and determined whether they relate to one object by calculating the included angle of two normal vectors. The former method focused on whole object segmentation, whereas the latter recognized as many connected edges as possible. To better exploit supervoxel features and their contextual relationships for point cloud classification, we propose a robust and effective classification approach that integrates point cloud supervoxels and their LCCP relations into an RF classifier to improve the accuracy of feature calculation and reduce computational costs.

## 3. Methodology

### 3.1. Overview of the Approach

The approach starts with a voxel-grid-based downsampling algorithm [39] to prevent the point cloud from becoming over-dense without impacting the original structure. Next, a noise-rejection statistical-outlier-removal filter is used to remove dynamic objects and erroneous points from the aerial laser point cloud. The threshold is calculated from the average distance between a single point and its k-neighbors referring to a certain range of standard deviation.

The technical route for our approach after data preprocessing is shown in Figure 1. The features are divided into three categories, point-based, eigen-based, and grid-based. First, the original 3D point cloud is transformed into a set of supervoxels by the supervoxel calculation method, in which points located in the same supervoxel generally have similar feature descriptions. The original point cloud is also divided using a regular grid to facilitate the extraction of grid-based elevation features in the later stage. Instead of semantic labeling of the raw points, supervoxels are used as the basic unit for semantic classification, and the centroids of the supervoxels are generated from the supervoxel structure. Three kinds of features are calculated: (1) The eigen-based features are first calculated using a principal component analysis algorithm, and the corresponding geometric shape features are generated by deformation and combination with those eigenvalues. Specifically, the adjacency relationship built by voxel cloud connectivity segmentation (VCCS) is used to determine the supervoxel neighborhood ranges. (2) The point-based features, including the local density, point feature histogram, point's normal vectors, elevation values, and RGB color properties, are obtained via neighborhood calculation or the point cloud's raw attributes. (3) We introduce a grid-based elevation feature to decrease the influence of uneven topography during point cloud classification. Based on the regularized grid of the point cloud data, the relative elevation of the horizontal location is used as the elevation feature of each supervoxel centroid. Finally, all three feature types are used to train the supervoxel-based RF model, which is used for point cloud classification.

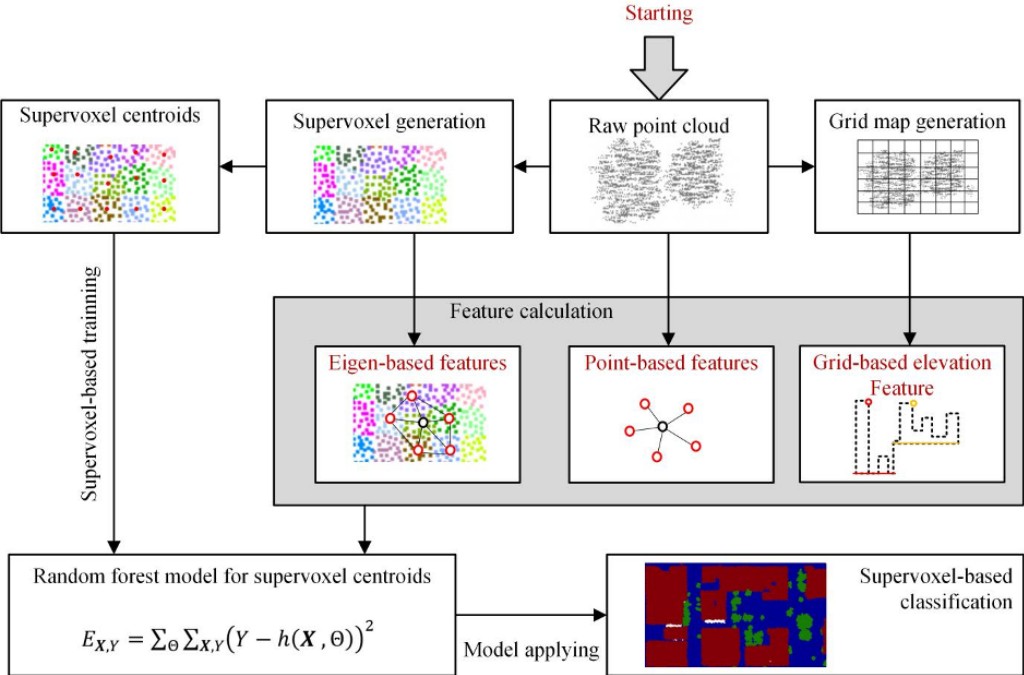

**Figure 1.** Supervoxel-based random forests framework for point cloud classification. The equation of the random forest model located at the bottom-left refers to the least squares method applied in the model to predict unlabeled points, in which *Y* represents the label, *X* represents an individual centroid point, and Θ represents the coefficient matrix.

### 3.2. Two-Level Graphical Model Generation for Feature Extraction

Supervoxels are defined as groups of points that contain similar geometric features or attributes, such as location, color, and normal direction. Additionally, adjacency relationships embedded in supervoxels can provide more effective information for neighborhood searching, improving the robustness and accuracy of feature calculation. For this classification method, we use supervoxels, rather than single points, as the basic unit to construct the RF model, and the domain information is constrained via LCCP segmentation. Therefore, a two-level graphical model using supervoxel calculation and LCCP optimization is generated from the raw point cloud. Figure 2 illustrates the two-level graphical model generation process.

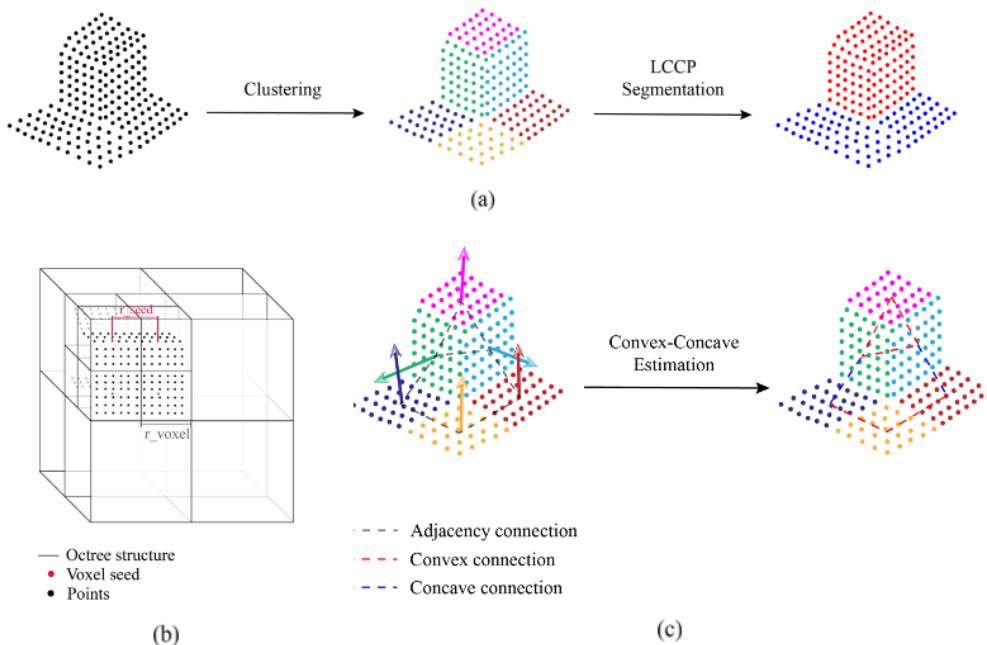

**Figure 2.** Illustration of two-level graphical model generation. (**a**) The fundamental process of supervoxel-based object segmentation. (**b**) The octree structure used for supervoxel clustering. (**c**) The locally convex connected patches (LCCP) segmentation scheme. Colored arrows show the corresponding normal vectors of supervoxels.

#### 3.2.1. First-Level Graphical Model Generation by the Supervoxel and VCCS Algorithm

First, we generate the supervoxel model in two steps, namely, randomly setting down seeds within the point cloud and clustering by calculating the feature distances among neighboring points. The supervoxel clustering algorithm estimates the point homogeneity via color, space, and normal dimensions as in Equation (1).

$$
\begin{aligned}
d &= i_{space} * d_{space} + i_{normal} * d_{normal} \\
d_{space} &= \frac{\sqrt{\Delta x^2 + \Delta y^2 + \Delta z^2}}{r_{voxel}} \\
d_{normal} &= \frac{v_1 * v_2}{\mid v_1 \mid * \mid v_2 \mid}
\end{aligned}
\tag{1}
$$

where $d$ represents the summarized estimation of homogeneity across all dimensions, $d_{space}$ represents the Euclidean distances between the seed points and surrounding points, and $d_{normal}$ is the normal of the fitted plane by the least squares fitting method based on the neighbor points. In this approach, the weights for distance $i_{space}$ and normal $i_{normal}$ during supervoxel clustering are set to 0.4 and 0.6, in which the higher the weight, the greater the contribution. $r_{voxel}$ is the size of each supervoxel, and $v_1$ and $v_2$ are the normal vectors

of pairwise adjacent supervoxels. The entire point cloud is clustered into supervoxels using Voxel Cloud Connectivity Segmentation (VCCS) as proposed by [18]. Figure 2b shows the schemes for supervoxel generation in which the octree structure is used to define branches and separate areas. Based on the supervoxel clustering results, the centroids of each supervoxel are calculated and then used for RF point cloud classification. All points within their respective supervoxel have similar features, and the centroid points are ordered in a mesh-like shape to simplify the complex computation of plane shape features. Specifically, an adjacency map containing the adjacent connections relations among supervoxels is simultaneously generated, which presents coterminous connection information that can greatly reduce the cost of neighborhood searching and improve the robustness and accuracy during neighbor calculation [40,41].

### 3.2.2. Second-Level Graphical Model Generation via LCCP Calculation

In order to determine the neighborhood relationship more accurately, we realize the extraction of a second-level graphical model by applying the Locally Connected Convex Patches (LCCP) algorithm on the first-level supervoxel model. In this algorithm, the connection relations implicit in the supervoxels are used for the determination of the neighborhood information, and these connection relations are defined as edges. The edges between adjacent supervoxels are given concave and convex type information based on a surface convexity detection. In order to ensure the aggregation of neighboring super voxels with similar characteristics, we calculate the "robust neighbors" of each supervoxel by judging the concave–convex relationship of edges. "Robust neighbors" means that the domain information can more reasonably represent the geometric features of the current location. Figure 2 shows the convex–concave estimation method among the supervoxels. The method of determining the concave–convex relationship is shown in Equation (2). When two super voxels have a concave domain relationship, they are considered to belong to two different objects. Therefore, after LCCP-based calculations, the adjacency relations of super voxels are given concave and convex properties, which can assist in obtaining more robust domain information quickly and accurately during feature calculations.

$$\hat{d} = \frac{\vec{x_1} - \vec{x_2}}{\| \vec{x_1} - \vec{x_2} \|}$$
$$\Delta\alpha = \vec{n_1} \cdot \hat{d} - \vec{n_2} \cdot \hat{d} \tag{2}$$

where $\vec{x_1}$ and $\vec{x_2}$ indicate the centroids of these two observed two supervoxels, and $\vec{n_1}$ and $\vec{n_2}$ represent their normal vectors. The relationship is considered a convex connection when $\Delta\alpha > 0$, which indicates the angle between the normal vector of the current supervoxel and the linear vector defined by $\vec{x_1} - \vec{x_2}$ is small. Alternatively, the relationship is considered a concave connection when $\Delta\alpha < 0$.

### *3.3. Hybrid Feature Description*

### 3.3.1. Point-Based Feature Description

Considering that some features are extracted from the original point cloud with better robustness, we present five types of point cloud feature description and extraction methods. The five main types contain "Local density", "Point feature histogram (PFH)", "Direction", "Relative elevation", and "RGB color", as follows.

(1) **Local density of the point cloud:** the density feature is calculated as the average distance from one point to the nearest k-neighbors. For each centroid in the super voxel, fast retrieval of domain points is achieved by the construction of a KDTREE and the fast library for approximate nearest neighbors (FLANN) algorithm [42]. Then, the local density feature of the point is obtained by calculating the average of the Euclidean distance between two pairs of neighboring points.

(2) **Point feature histogram (PFH):** The goal of the PFH formulation is to encode a point's k-neighborhood geometrical properties by generalizing the mean curvature around

the point using a multidimensional histogram of values [43,44]. A Point Feature Histogram representation is based on the relationships between the points in the k-neighborhood and their estimated surface normals. In this work, the PFH feature of each centroid point is calculated by KDTREE searching from the original point cloud.

(3) **Direction:** The direction feature indicates the angle between the normal of the location and the horizontal plane, which is calculated as follows (Equation (3)).

$$c = \frac{n_1 \cdot n_2}{\mid n_1 \mid \cdot \mid n_2 \mid} = \frac{z_1}{\sqrt{x_1{}^2 + y_1{}^2 + z_1{}^2}} \tag{3}$$

where $c$ refers to the cosine value, $n_1$ represents the normal vector of the supervoxel, and $n_2$ is the normal vector of the horizontal plane (defined as (0,0,1)), respectively. In this paper, to facilitate feature normalization, the cosine value is used to represent the directional features of the supervoxels.

(4) **Relative elevation:** The relative elevation feature is the distance from the center point of the supervoxel to the ground in the extended z-direction. Considering the influence of ground undulation on elevation features, this paper proposes a grid-based optimization strategy for elevation feature extraction (see Section 3.3.3).

(5) **RGB color:** RGB color information can achieve effective judgment of feature types, and this paper uses color features as a basic feature of supervoxels. Considering that this paper uses supervoxels as the basic unit for feature classification experiments, their color features are determined by the average value of points inside the supervoxels.

### 3.3.2. Eigen-Based Feature Description

Eigen values illustrate the local shape characteristics of the point neighborhood, which helps distinguish objects, such as ground points which have small values in one direction and vegetation points which have similar values. The traditional method of computing Eigen-based features is implemented by K-neighborhood search of point clouds. In order to obtain more robust neighborhood information, this paper implements accurate neighborhood estimation based on the LCCP algorithm, which can accurately estimate the boundaries of different types of objects. Then these neighborhood supervoxels satisfying the LCCP conditions are used for Eigen-based feature calculation. Figure 3 shows the flow of the super voxel neighborhood calculation method in which the concave–convex relationship between supervoxels is derived from the second-level graphic model.

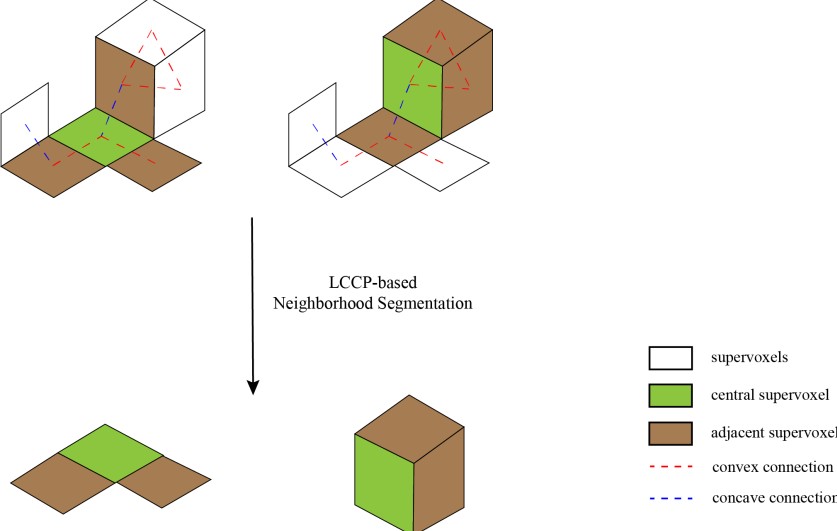

**Figure 3.** Locally convex connected patches (LCCP) neighborhood optimization. The neighborhood ranges used to calculate eigenvalues are shown at the bottom.

The three eigenvalues will be calculated by feature decomposition, and sorted in descending order ($\lambda_1 \geq \lambda_2 \geq \lambda_3 \geq 0$). Based on the mathematical meaning of eigenvalues, different combinations of eigenvalues demonstrate particular shape characteristics [10]. In this work, five types of shape features are used for the classification of supervoxels, including "Curvature", "Linearity", "Planarity", "Scattering", and "Anisotropy". The specific calculation formulas are shown in Table 1.

(1)　Curvature: Describes the extent of the curve for a point group.
(2)　Linearity: Describes the extent of the line-like shape for a point group.
(3)　Planarity: Describes the extent of the plane-like shape for a point group.
(4)　Scattering: Describes the extent of the sphere-like shape for a point group.
(5)　Anisotropy: Describes the difference between the extents of entropy in respective directions of eigenvectors for a point group.

**Table 1.** Computing method for eigenvalue-based shape features. Feature definitions on the left are described in Section 3.3.2. Three eigenvalue symbols are sorted in descending order from 1 to 3 in the formulas.

| Feature Definition | Computing Formula |
|:---:|:---:|
| Curvature | $\mathcal{C}_e = \frac{\lambda_3}{\lambda_1 + \lambda_2 + \lambda_3}$ |
| Linearity | $\mathcal{L}_e = \frac{\lambda_1 - \lambda_2}{\lambda_1}$ |
| Planarity | $\mathcal{P}_e = \frac{\lambda_2 - \lambda_3}{\lambda_1}$ |
| Scattering | $\mathcal{S}_e = \frac{\lambda_1}{\lambda_3}$ |
| Anisotropy | $\mathcal{A}_e = \frac{\lambda_3 - \lambda_1}{\lambda_3}$ |

### 3.3.3. Grid-Based Elevation Feature Description

When the original elevation features of point cloud data are used for point cloud data classification, it is easy to produce misclassification in areas with large topographic undulations. In particular, features with similar geometric shapes or colors can easily cause confusion in classification, such as the ground and the top surfaces of buildings. Some methods use DEM information to reduce the influence of terrain height difference on data classification, but it is often difficult to obtain accurate DEM data. Therefore, this paper proposes a grid-based method for calculating elevation features, which can accurately calculate the relative elevation information between the features and the ground. As shown in Figure 4a, we first project the original point cloud data onto a 2D plane, i.e., XOY plane and then divide the projected data into a grid according to the area size. Therefore, the relative elevation of each point can be obtained by subtracting the ground elevation from that point. In general, we take the smallest elevation value in the grid as the ground elevation of the target location. However, some hindrances, such as the absence of ground points below the building roof and large-scale clusters, are typical in 3D urban scenes due to the shortage of ray reflection, meaning that roof points, especially with a flat shape, are occasionally confused with ground points. A lattice filter kernel is used to solve the ground detection error problem, the basic principle of which is similar to image processing [45]. As illustrated in Figure 4b, each cell is checked by a $5 \times 5$ filter kernel, the outliers are first removed by the Gaussian distribution strategy. Then the algorithm corrects the ground elevation value of the current cell with the average value of the filter, when the standard deviation does not satisfy the Gaussian distribution condition [46].

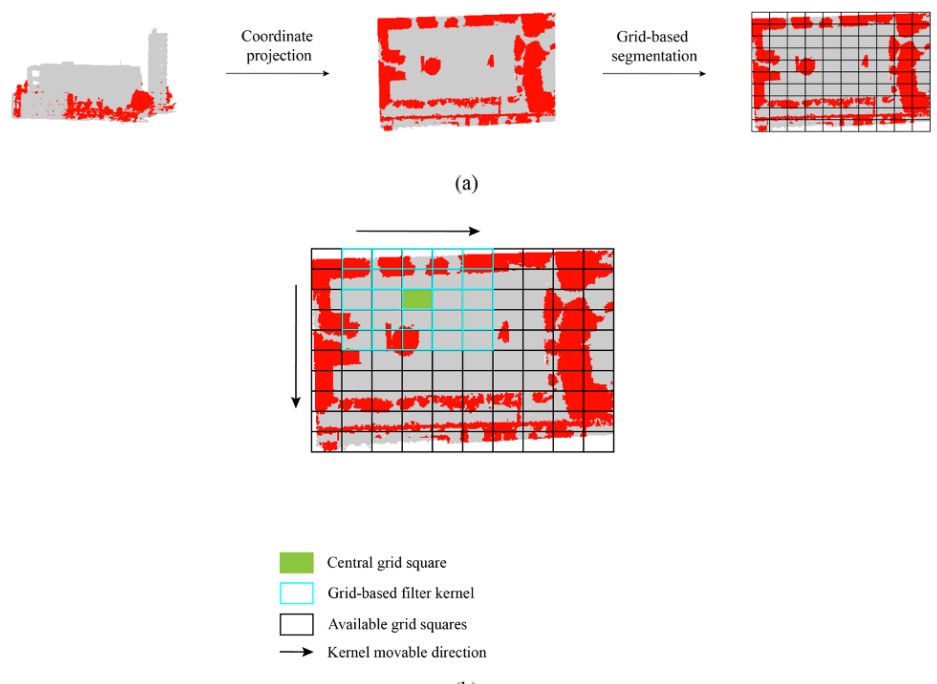

**Figure 4.** Grid-based elevation computation and filtering. (**a**) The illustrated point cloud data (left) and the 2D-projected data with grid segmentation (right). (**b**) The grid filter examining anomalies of calculated elevation values in grid squares.

### 3.4. Supervoxel-Based Random Forests (RF) Model

The RF model is an ensemble learning method for classification, regression, and other tasks that operates by constructing a multitude of decision trees at training time. For classification tasks, the output of the random forest is the class selected by the most trees. In order to integrate the above three hybrid features for point cloud data classification, a supervoxel-based RF model is constructed in this paper. In this method, the supervoxels will be used as the basic classification units, and the extracted hybrid features will be used as training information input for decision tree generation. The random forest construction process is constrained by two main parameters including the "max depth" and the "total number of decision trees". Here, the "max depth" represents the depth of each tree in the forest. The deeper the tree, the more splits it has, and it captures more information about the data. However, too large a depth value can easily cause problems such as overfitting or excess processing time. In this paper, to balance operational efficiency and classification accuracy, the max depth and the total number of decision trees are set to 25 and 10, respectively. So to obtain an optimal number, the accuracy of the output RF model is verified with the validation set. The algorithm applies the mean squared generalization error to evaluate the classification correctness, as Equation (4) shown in [20].

$$E_{X,Y} = \Sigma_\Theta \Sigma_{X,Y} (Y - h(X, \Theta))^2 \tag{4}$$

where $X$ refers to the random feature vector, and $Y$ refers to the corresponding label. $\Theta$ is a single tree inside the forest, appearing in tandem with one $X$.

The framework proposed by the ETH Zurich RF template library [47] is used to train the supervoxel-based random forest model. It should be noted that the framework contains three kinds of classification method, including ordinary classification, local smooth classification, and graph cut-based classification. In our approach, graph cut-based classification is employed for training purposes, since it is optimized with an energy minimization method [48] and provides the best overall classification accuracy.

## 4. Experimental Results

To verify the effectiveness of the proposed method in this paper, two sets of data were used for classification testing and accuracy analysis. The publicly available dataset from the ISPRS benchmark [49] contains data collected in Toronto, Canada, and Vaihingen, Germany, both the Toronto and German datasets were used for accuracy verification. Subsequently, a classification experiment was conducted with the airborne LiDAR dataset collected in Shenzhen City, China. In our experiments, three accuracy assessment metrics were used for accuracy evaluation according to the conventional accuracy assessment methods for point cloud classification [50,51]. We selected three indices, including the overall accuracy (OA), the mean intersection over union (mIoU), and the F1-score, which were calculated as follows.

$$OA = \frac{True\ Positive}{True\ Positive + True\ Negative + False\ Positive + False\ Negative}$$

$$mIoU = \frac{True\ Positive}{True\ Positive + 2 \times (True\ Negative + False\ Positive + False\ Negative)}$$

$$p = \frac{True\ Positive}{True\ Positive + False\ Positive} \quad (5)$$

$$r = \frac{True\ Positive}{True\ Positive + False\ Negative}$$

$$F_1 = \frac{2 * p * r}{p + r}$$

where *True Positive* (TP), *False Positive* (FP), *True Negative* (TN), and *False Negative* (FN) values are extracted from the confusion matrix of the classification result, and *p* and *r* are the precision and recall percentages, respectively.

### 4.1. ISPRS Benchmark Datasets

#### 4.1.1. Toronto Sites

The Toronto dataset was divided into two regions, Area 1 and Area 2 for testing purposes. The classification results are shown in Figure 5. The overall scene was divided into four types, buildings, vegetation, ground, and background. As shown in Figure 5, there was a large amount of overlap and crossover between buildings and vegetation in the Toronto data, as well as incomplete facade collection, which can easily lead to the problem of confusion between tree and building facades during classification process. Meanwhile, due to the lack of color information in Toronto's point cloud data, the classifier relied more on geometric features for semantic classification.Thanks to the grid-based elevation features and the supervoxel-optimized Eigen features, the proposed algorithm still achieved good classification results when only geometric features were used. Figure 6 shows the comparison of the classification accuracy before and after using the grid-based elevation features, in which it can be clearly seen that the ground level and the top surfaces of the buildings could be accurately distinguished after the optimization of the elevation features.

However, the method proposed in this paper still suffered from some classification errors. As illustrated in Figure 7, some misclassified areas are shown enlarged; those errors were mainly caused by similar geometric features or missing data. For example, some buildings were incorrectly classified as ground due to their low elevation values, and some buildings with missing facades were classified as ground.

In addition, the quantitative classification results were compared with those of five state-of-the-art algorithms, including MAR_2, MSR, ITCM, TICR, and TUM. The first two rely mainly on the geometric information of the original point cloud for classification, while the last three approaches fuse point cloud and image features for classification.The OA, mIoU, and F1-score are listed in Table 2. The proposed method achieved high accuracy classification results in both regions, similar to the classification accuracy of MAR_2 and MAR. It should be noted that the MSR method achieved better classification accuracy in most cases, mainly

due to the use of DEM data. The proposed method achieved an OA accuracy of 93.2%, mIoU accuracy of 87.4%, and an F1-score of 92.6% in Area 1; in particular, the F1 accuracy was the best among all methods. Similarly, in Area 2, the classification method proposed in this paper achieved an OA accuracy of 93.1%, an mIoU accuracy of 87%, and an F1_score of 85.8% respectively.

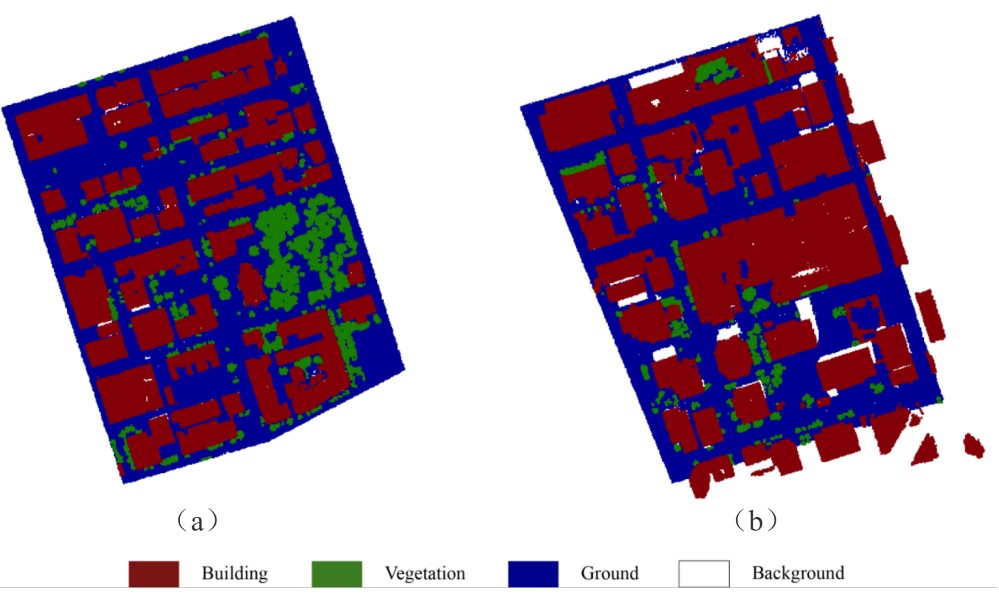

**Figure 5.** Classification results of two Toronto site areas. (**a**) The classification result of Area 1 and (**b**) the clasification result of Area 2.

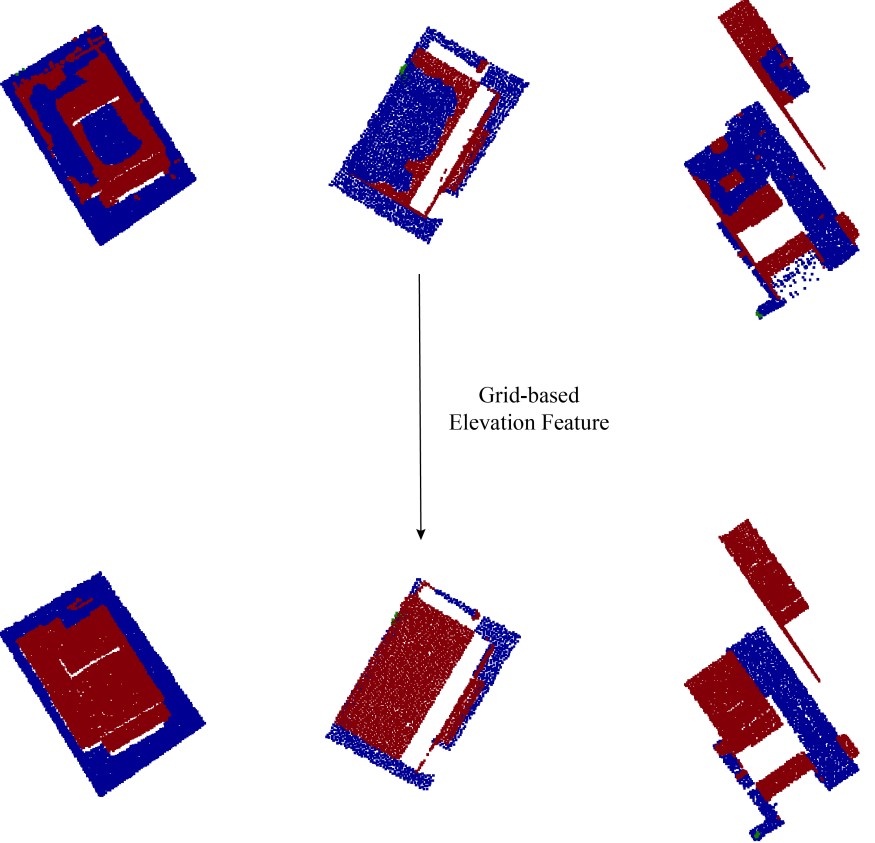

**Figure 6.** The comparison of the classification accuracy before and after using the grid-based elevation features on the Toronto sites.

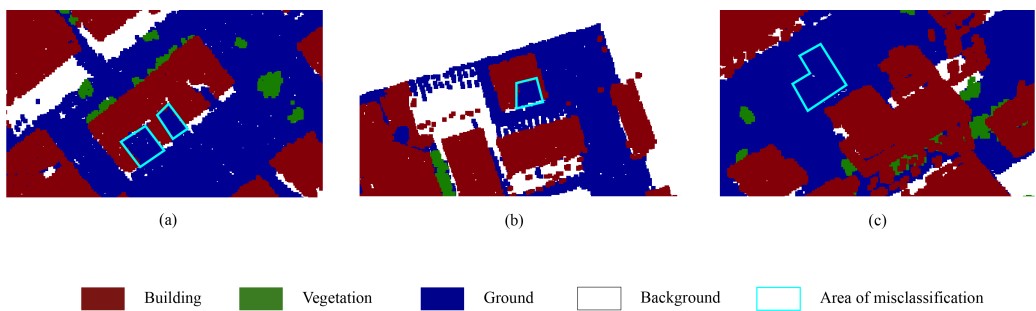

**Figure 7.** Misclassification cases in which roof points were recognized as ground points in the Toronto sites. (**a**–**c**) refer to different types of misclassification results from roof to ground separately.

**Table 2.** Quantitative comparison of the proposed method and previous related methods tested on the Toronto sites. Two methods, MAR_2 and MSR, used only the point cloud for classification; MSR applied terrestrial digital models. ITCM, ITCR, and TUM used the point cloud and images.

| Methods | Area 1 | | | Area 2 | | |
|---|---|---|---|---|---|---|
| | OA (%) | mIoU (%) | F1-Score (%) | OA (%) | mIoU (%) | F1-Score (%) |
| MAR_2 | 94.3 | 89.2 | 88.9 | 94.0 | 88.7 | 88.4 |
| MSR | 95.5 | 91.4 | 91.2 | 94.8 | 90.1 | 89.7 |
| ITCM | 81.3 | 68.5 | 66.1 | 83.0 | 70.9 | 67.9 |
| ITCR | 84.2 | 72.7 | 69.2 | 85.4 | 74.5 | 72.4 |
| TUM | 82.6 | 70.4 | 68.1 | 83.1 | 71.1 | 68.9 |
| Our method | 93.2 | 87.4 | 92.6 | 93.1 | 87.0 | 85.8 |

OA, overall accuracy; mIoU, mean intersection over union.

### 4.1.2. Vaihingen Sites

The height of buildings in the Vaihingen data was similar to the vegetation and did not contain color information, which was be a major challenge for point cloud data classification for the data set. Similar to the experiment of the Toronto area, the scene was divided into four categories of labels, buildings, vegetation, ground, and background. The classification results are shown in Figure 8. It can be clearly seen that the classification results were worse than those of the Toronto data, which was mainly caused by the similarity of geometric features among different types. Due to connections between supervoxels containing medium-height vegetation and building facades and some oddly curved roof surfaces, points with building groundtruth values were more likely to be partially or completely misjudged as trees. Figure 9 shows some cases of misclassification in the Vaihingen region, in which some parts of buildings were misclassified into trees.

Meanwhile, seven existing classification algorithms were used for comparative analysis of classification accuracy. The OA, mIoU, and F1-score are listed in Table 3. It can be seen that the classification algorithm proposed in this paper achieved the best classification accuracy of 85.2% OA, 74.2% mIoU accuracy, and 83.6% F1_score accuracy, respectively.

However, with the building outline explicitly extracted, the proposed method performed well in the remaining areas, achieving an overall F1-score above 83%, which surpassed some methods using heterogeneous data sources.

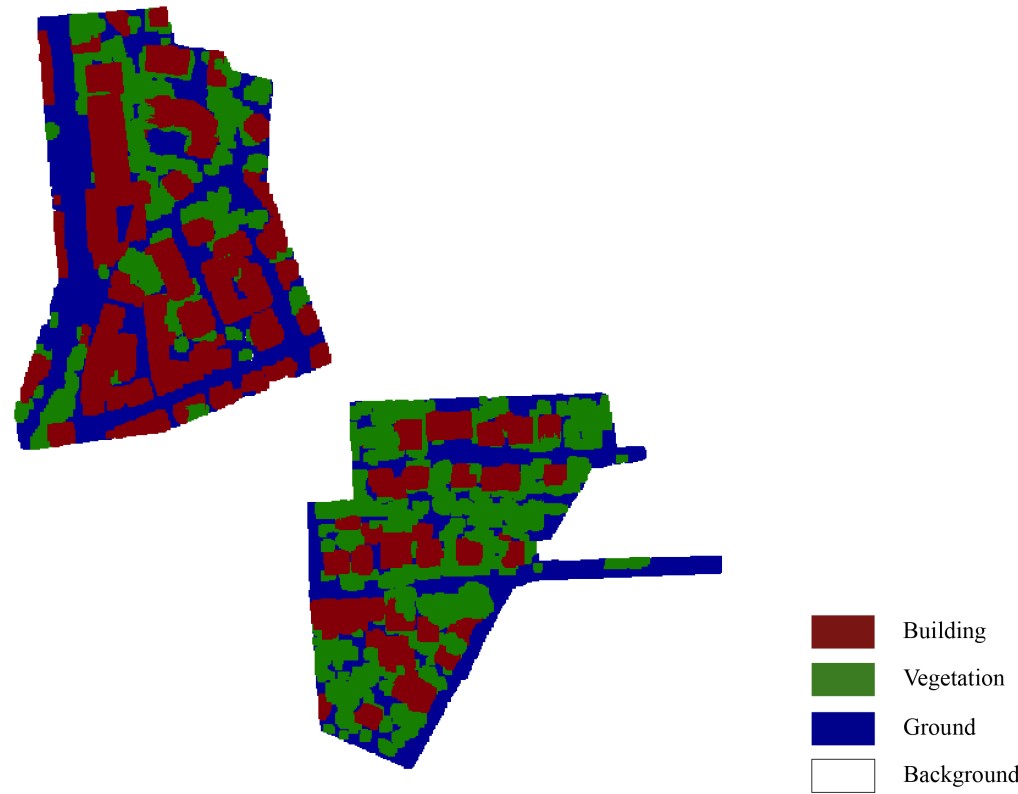

**Figure 8.** Classification results of the Vaihingen sites.

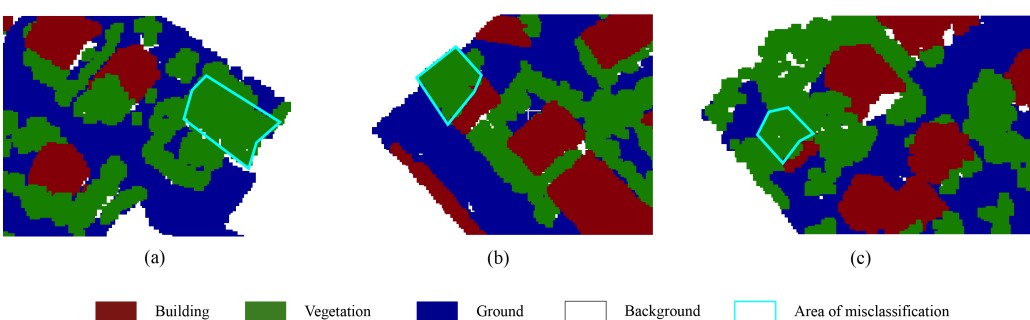

**Figure 9.** Misclassified regions in the Vaihingen site caused by unexpected connections between supervoxels of different objects. (**a–c**) mean misclassification situations in different minor scenes from roof to vegetation.

**Table 3.** Quantitative comparison of the proposed method and previous related methods tested on the Vaihingen sites sorted by overall accuracy (OA) in ascending order. The F1-score was computed based on the same categories (building, vegetation, and ground).

| Methods | OA (%) | mIoU (%) | F1-Score (%) |
|---|---|---|---|
| UM | 80.8 | 67.8 | 78.1 |
| BIJ_W | 81.5 | 68.8 | 78.6 |
| LUH | 81.6 | 68.9 | 80.4 |
| RIT_1 | 81.6 | 68.9 | 79.0 |
| D_FCN | 82.2 | 69.8 | 80.9 |
| WhuY3 | 82.3 | 69.9 | 81.0 |
| WhuY4 | 84.9 | 73.8 | 80.8 |
| Our method | 85.2 | 74.2 | 83.6 |

mIoU, mean intersection over union.

*4.2. Airborne Laser Scanner Dataset in Urban Scenes of Shenzhen*

RGB color information plays a significant role in the proposed classifier because three discriminative features are computed by RGB reflection data, and multispectral aerial images cannot be included. Furthermore, the two datasets used for testing carried little or incomplete spectral band information. Point cloud data assisted by spectral information during generation and reconstruction with complete color data and high resolution can more comprehensively prove the performance of the proposed method. Integrated reconstruction of the facade is also beneficial for the extraction of buildings.

The selected dataset included four urban regions, one for the training set and three for independent validation [marked as (a), (b), (c)]. The training area was 350 m × 200 m, and the validation areas were approximately 400 m × 300 m. The entire dataset was downsampled to a resolution of 0.3 m. The classification results are illustrated in Figure 10. Most vegetation points and ground points were accurately classified, and explicit outlines of buildings were visible in the resulting figure. In most scenes, vegetation was distinguished from adjacent buildings. Moreover, the centroid-based classification method enabled low computation costs, even though each validation area contained more than four million points after the downsampling process. This demonstrates that the proposed classifier successfully handles large datasets. The point-based classification method in CGAL library [52] was used for comparison purpose. The quantitative performance evaluations of our proposed method and the pointbased method are shown in Table 4. As expected, the super voxel-based method proposed in this paper achieved better classification accuracy in all three regions compared to the traditional point cloud-based methods. Specifically, the proposed method achieved 3.6, 5.8, and 4.4 percent, respectively, in the OA, mIoU, and F1_score in Area (a). Similar results were found in the other two regions.

The average performance of the proposed method was higher for the Shenzhen dataset than the Vaihingen and Toronto datasets. The mostly rectangular rooftop shapes and integrated facade structures prevented building points from being recognized as vegetation, whereas the uncertainty of object consistency in the Vaihingen set led to false classification. Compared with the Toronto sites, which were comparably generated except without color information, most elevated vegetation points and buildings with low height and more detailed facades were successfully distinguished using RGB color features in the Shenzhen dataset. However, some exceptional situations in the dataset affected the overall accuracy of the classification results. As shown in Figure 11a, the neighborhood information of partial rooftop points that were similar to roads, such as rises at the edge or street light posts, reduced the contextual consistency of the local region and affected the classification. Additionally, due to the intricate and uncertain shape appearances in modern urban scenes, a single training area provided limited polygonal examples. Parts of buildings with minor scale or unusual contours that were not provided in the training region were misclassified as ground pieces in the validation sets [Figure 11b], which reduced the overall classification accuracy.

Benefiting from supervoxel extraction processing, the point cloud of Shenzhen University can be rapidly aggregated into supervoxel structures, which effectively reduced the point cloud density and complexity. In turn, with supervoxels as the basic unit, the classification method proposed in this paper achieved point cloud classification with high efficiency, and the overall computation costs were about 1.5 h. Moreover, the utilization of LCCP object homogeneity segmentation in supervoxel-based neighborhoods contributed to the considerable classification precision with complete object surfaces consisting of point arrays, which advanced the object-based theory.

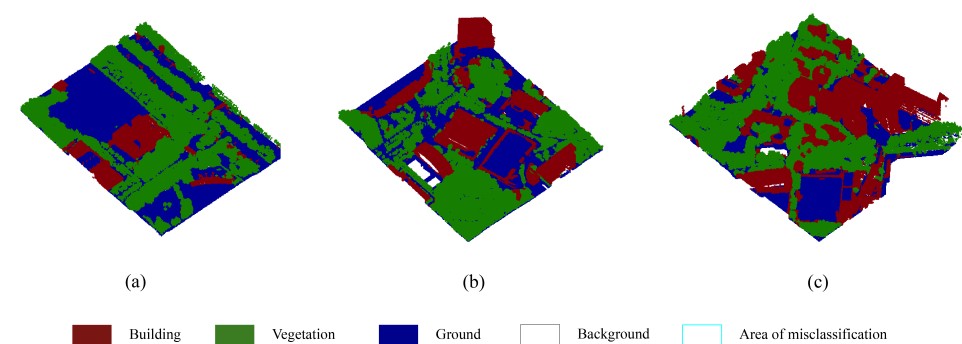

**Figure 10.** Classification results of airborne LiDAR-generated Shenzhen sites. Three selected sites have been marked as (**a**–**c**).

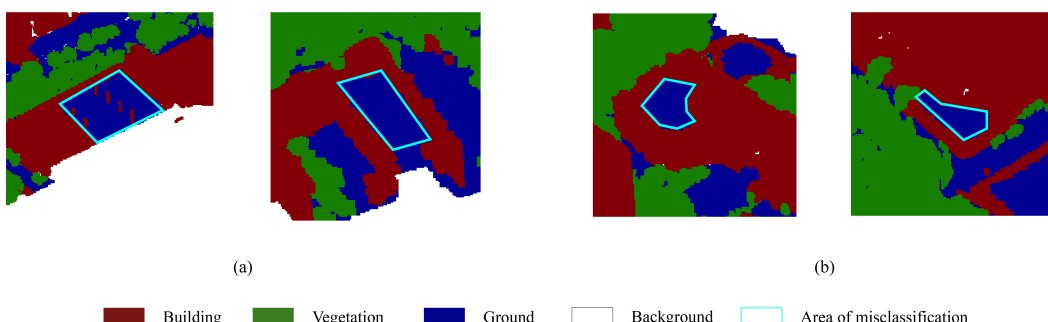

**Figure 11.** Misclassification cases in the Shenzhen dataset. (**a**) Faults due to edge interruption. (**b**) Faults due to untrained object shapes.

**Table 4.** Quantitative evaluation of the supervoxel-based results and point-based results of the proposed method on the Shenzhen airborne LiDAR dataset.

|  | Area (a) | | | Area (b) | | | Area (c) | | |
|---|---|---|---|---|---|---|---|---|---|
|  | OA (%) | mIoU (%) | F1-Score (%) | OA (%) | mIoU (%) | F1-Score (%) | OA (%) | mIoU (%) | F1-Score (%) |
| Our method | 94.0 | 88.7 | 90.1 | 93.5 | 87.8 | 91.8 | 93.5 | 87.8 | 91.7 |
| Point based | 90.6 | 82.9 | 85.6 | 87.6 | 78.0 | 84.2 | 86.1 | 75.6 | 79.8 |

OA, overall accuracy; mIoU, mean intersection over union.

### 4.3. Discussions of the Experimental Results

For the classification results of the ISPRS benchmark datasets, due to missing RGB color information and some incomplete facades of buildings, the classifier lacked RGB band features, and eigen features were less discriminative. As a result, separated low roofs were classified as vegetation with a similar height. However, most of the borders dividing buildings and vegetation were successfully detected, which showed the excellent effect of applying VCCS and LCCP object-based segmentation into the classifier. For the result of the dataset of the Shenzhen urban scene, although complicated urban scenes provided multi-aspect obstacles for the classifier, the outcome of the proposed method reached our expectations. The proposed classifier achieved a high accuracy classification using only 3D point cloud data without the assistance of digital models and multispectral images, as illustrated in the ISPRS benchmark site outputs. Furthermore, benefited by the RGB information contained in this dataset, the borders between two objects in different types were more distinct, which means color information assisted the object-based classification process.

### 5. Conclusions

In this paper, we proposed a robust and effective airborne LiDAR point cloud classification method that integrated hybrid features, including point-based features, eigen-based

features, and elevation-based features, into a supervoxel RF model. Three main innovations were applied to effectively improve the classification accuracy of the proposed model.

(1) Rather than single points, we used supervoxels as the basic entity to construct the RF model and constrain the domain information via LCCP segmentation.

(2) A two-level graphical model involving supervoxel calculation and LCCP optimization was generated from the raw point cloud, which significantly improved the reliability and accuracy of neighborhood searching.

(3) The features were divided into three categories based on feature descriptions (point-based, eigen-based, and grid-based), and three unique feature calculation strategies were accordingly designed to improve feature reliability. We conducted three experiments using ALS data provided by ISPRS and real scene data collected from Shenzhen, China, respectively. We compared the quantitative analysis of ALS datasets with other state-of-the-art methods, and the classification results demonstrated the robustness and effectiveness of the proposed method. Furthermore, this method achieved fine-scale classification when the point clouds had different densities.

However, the proposed method still had some limitations on scene generalizability. The algorithm may fail to recognize roof components when lacking facade information, which is caused by a loss of the connection relationship between supervoxels. In the future, we would like to integrate external constraints into the classification process to prevent the influence of over-segmentation.

**Author Contributions:** Data curation, J.L.; Formal analysis, L.L.; Funding acquisition, W.W.; Investigation, R.G.; Methodology, L.L. and S.T.; Project administration, S.T.; Supervision, S.T. and R.G.; Validation, X.L.; Visualization, Y.L.; Writing—original draft, L.L.; Writing—review & editing, S.T. All authors have read and agreed to the published version of the manuscript.

**Funding:** This research received no external funding.

**Institutional Review Board Statement:** Not applicable.

**Informed Consent Statement:** Not applicable.

**Data Availability Statement:** Not applicable.

**Acknowledgments:** This work was supported in part by the National Key Research and Development Program of China (Projects Nos. 2019YFB210310, 2019YFB2103104) and in part by a Research Program of Shenzhen S and T Innovation Committee grant (Projects Nos. JCYJ20210324093012033, JCYJ20210324093600002), the Natural Science Foundation of Guangdong Province grant (Projects No. 2121A1515012574), the Open Fund of Key Laboratory of Urban Land Resources Monitoring and Simulation, MNR (Nos. KF-2021-06-125, KF-2019-04-014), the National Natural Science Foundation of China grant (Projects Nos. 71901147, 41901329, 41971354, 41971341) and the Foshan City to promote scientific and technological achievements of universities to serve industrial development support projects(Projects No. 2020DZXX04).

**Conflicts of Interest:** The authors declare no conflict of interest.

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
