# Peer review of "A Supervoxel-Based Random Forest Method for Robust and Effective Airborne LiDAR Point Cloud Classification"

_remotesensing, doi:10.3390/rs14061516_

Round 1

Reviewer 1 Report

The paper describes with sufficient detail the methodology, and presents the experimental results of a LiDAR point cloud classification method that integrates point cloud supervoxels and their locally convex connected patches into a Random Forest classifier.

The structure of the paper is good. A brief but comprehensive reference to the knowledge-driven and model-driven classification approaches is given; the description of the methodology of the proposed approach follows. The approach uses existing techniques, but innovations have been added to the process, such as the use of supervoxels instead of single points as the basic entities to construct the model and the deviation of the features into three categories of feature descriptions (point-based, eigen-based, and grid- based). The presentation of the methodology is satisfactory and the accompanying figures are explanatory.

From the development of case studies arise the limitations of the proposed approach and some classification errors that indicate the need for further improvement of the method.

Additions that must be made in the chapter ‘Experimental results’, are:

-  specific data on the computation costs in the application of the proposed approach (the computation costs referred only as ‘acceptable’)

-  results from the use of other existing classification algorithms with the Shenzhen dataset, in order to have a comprehensive comparative analysis of the classification accuracy (as in the cases of Vaihingen and Toronto datasets).

Author Response

Response to Review Comments

A SUPERVOXEL-BASED RANDOM FORESTS CLASSIFIER FOR AIRBORNE LASER SCANNER 3D POINT CLOUDS

(First of all, we would like to express our sincere appreciation to the comments and suggestions from the reviewers!)

Review #1:
The paper describes with sufficient detail the methodology, and presents the experimental results of a LiDAR point cloud classification method that integrates point cloud supervoxels and their locally convex connected patches into a Random Forest classifier.The structure of the paper is good. A brief but comprehensive reference to the knowledge-driven and model-driven classification approaches is given; the description of the methodology of the proposed approach follows. The approach uses existing techniques, but innovations have been added to the process, such as the use of supervoxels instead of single points as the basic entities to construct the model and the deviation of the features into three categories of feature descriptions (point-based, eigen-based, and grid- based). The presentation of the methodology is satisfactory and the accompanying figures are explanatory.From the development of case studies arise the limitations of the proposed approach and some classification errors that indicate the need for further improvement of the method.Additions that must be made in the chapter ‘Experimental results’, are:

1.In the chapter ‘Experimental Results’, additions of specific data on the computation costs in the application of the proposed approach must be made (the computation costs referred only as 'acceptable")

Response: Thanks for pointing out this. The corresponding description of the computation costs are added in Section 3.2 marked in red text.
‘Benefiting from super voxel extraction processing, the point cloud of shenzhen university can be rapidly aggregated into super voxel structures and effectively reduce the point cloud density and complexity. In turn, with super voxels as the basic unit, the classification method proposed in this paper can achieve point cloud classification with high efficiency and the overall computation costs are about 1.5 hours.’

2.In the chapter ‘Experimental Results’, additions of results from the use of other existing classification algorithms with the Shenzhen dataset must be made, in order to have a comprehensive comparative analysis of the classification accuracy (as in the cases of Vaihingen and Toronto datasets).

Response: Thanks for the suggestion. The point-based classification method in CGAL library  is used for comparison purpose and the experimental results are shown in Table 4. The corresponding descriptions in terms of OA, mIoU and F1-score are added in Section 3.2

‘Table 4. Quantitative evaluation of the supervoxel-based results and point-based results of the proposed method on the Shenzhen airborne LiDAR dataset.
 Area (a) Area (b) Area (c)
 OA(%) mIoU(%) F1-score(%) OA(%) mIoU(%) F1-score(%) OA(%) mIoU(%) F1-score(%)
Our Method 94.0 88.7 90.1 93.5 87.8 91.8 93.5 87.8 91.7
Point based  90.6 82.9 85.6 87.6 78.0 84.2 86.1 75.6 79.8
OA, overall accuracy; mIoU, mean intersection over union; SV, supervoxel. 

The point-based classification method in CGAL library is used for comparison purpose. The quantitative performance evaluations of our proposed method and the point\-based method are shown in Table 4. As expected, the super voxel-based method proposed in this paper achieve better classification accuracy in all three regions compared to the traditional point cloud-based methods. Specifically, the proposed method achieve 3.6 , 5.8 and 4.4 percentage points respective in OA, mIoU, and F1_score in Area(a). The similar results can be found in other two regions.’

Reviewer 2 Report

The paper presents an interesting classification approach by supervoxels in the frame of the point cloud processing. The conclusions seem robust and consistent with the results. 
Nevertheless, I consider that some issues should be solved before accepting the manuscript. First, I have missed a separate subsection regarding Discussion or Concluding remarks in the (Experimental) Results section.

Regarding the way of citing, for instance, I would prefer "Huang et al. [22] integrated..." or "Wang et al. [31] developed" rather than "[22] integrated... " or "[31] developed". In addition, the sentence starting in Line 86 should be rewritten to be well understood.

There are no few typos (line 16, line 121, line 326 )or mistakes (we realizes), as a small sample. Please, check the article usage too.

Author Response

A SUPERVOXEL-BASED RANDOM FORESTS CLASSIFIER FOR AIRBORNE LASER SCANNER 3D POINT CLOUDS

(First of all, we would like to express our sincere appreciation to the comments and suggestions from the reviewers!)

Review #2:

The paper presents an interesting classification approach by supervoxels in the frame of the point cloud processing. The conclusions seem robust and consistent with the results. Nevertheless, I consider that some issues should be solved before accepting the manuscript.

  1. I have missed a separate subsection regarding Discussion or Concluding remarks in the (Experimental) Results section.

Response: Thanks for pointing out this.  As the concluding remarks of the experiments are discussed in their respective content,we do not describe it as a separate subsection in the original manuscript. According to your suggestion, all discussions in terms of the public datasets and the dataset of shenzhen are consolidated into Section 3.3 as follows.

3.3. Discussions of the experimental results

For the classification results of ISPRS benchmark datasets, due to missing of RGB color information and some incomplete facade part of buildings, the classifier lacks RGB band features and eigen features have become less discriminative. As a result, separated low roofs pretend to be classified as vegetations with a similar height. However, most of the borders dividing building and vegetation are successfully detected, which declares the excellent effect of applying VCCS and LCCP object-based segmentation into the classifier. For the result of the dataset of Shenzhen urban scene, although complicated urban scenes provide multi-aspect obstacles for the classifier, the outcome of the proposed method reaches our expectations. The proposed classifier achieves high accuracy classification using only 3D point cloud data without the assistance of digital models and multispectral images, as illustrated in the ISPRS benchmark site outputs. Furthermore, benefited by RGB information contained in this dataset, borders between two objects in different types behave more distinct, which means color information assisted the object-based classification process.

  1. Regarding the way of citing, for instance, I would prefer "Huang et al. [22] integrated… " or "Wang et al. [31] developed" rather than "[22] integrated... " or "[31] developed".

Response: All citation in this manuscript were checked and corrected.

  1. The sentence starting in Line 86 should be rewritten to be well understood.

Response: The corresponding content has been rewritten as follows.

For instance, Zheng et al. [24] used Fourier fitting method [25] to classify the pointcloud, in which the geometrical eigen features and basic features are integrated in their classification algorithm.

  1. There are no few typos (line 16, line 121, line 326 ) or mistakes (We realizes), as a small sample. Please, check the article usage too.

Response: The manuscript has been checked overall and all typos have been corrected.
